# The Plants of the *Asteraceae* Family as Agents in the Protection of Human Health

**DOI:** 10.3390/ijms22063009

**Published:** 2021-03-16

**Authors:** Agata Rolnik, Beata Olas

**Affiliations:** Department of General Biochemistry, Biology and Environmental Protection, University of Lodz, 90-236 Lodz, Poland; agata.rolnik@edu.uni.lodz.pl

**Keywords:** *Asteraceae* family, human health, antioxidant activity

## Abstract

The *Asteraceae* family is one of the largest flowering plant families, with over 1600 genera and 2500 species worldwide. Some of its most well-known taxa are lettuce, chicory, artichoke, daisy and dandelion. The members of the *Asteraceae* have been used in the diet and for medicine for centuries. Despite their wide diversity, most family members share a similar chemical composition: for example, all species are good sources of inulin, a natural polysaccharide with strong prebiotic properties. They also demonstrate strong antioxidant, anti-inflammatory and antimicrobial activity, as well as diuretic and wound healing properties. Their pharmacological effects can be attributed to their range of phytochemical compounds, including polyphenols, phenolic acids, flavonoids, acetylenes and triterpenes. One such example is arctiin: a ligand with numerous antioxidant, antiproliferative and desmutagenic activities. The family is also a source of sesquiterpene lactones: the secondary metabolites responsible for the bitter taste of many plants. This mini review examines the current state of literature regarding the positive effect of the *Asteraceae* family on human health.

## 1. Introduction 

Plants have long played a crucial role in the development of medicine, primarily due to their ability to synthesize secondary metabolites with potentially significant biological activity. In traditional medicine, plants were used in various ways to treat many different ailments. According to the World Health Organization, over 80% of the global population still depends on traditional and folk medicine, most of which is based on plant remedies. Drugs based on plants used in traditional medicine are often cheaper than normal drugs, are easily accessible and have fewer side effects than their synthetic alternatives. Many traditional medicinal plants have recently been analyzed using more modern methods, leading to the discovery of many promising compounds. These plant-derived compounds can be used in the modification of existing drugs or the design of completely new ones [1,2]. 

The majority of *Asteraceae* family members have therapeutic applications, and have a long history in traditional medicine: some members have been cultivated for more than 3000 years for edible and medical purposes. They are most common in arid and semi-arid regions of subtropical areas, but are known and distributed throughout the word. The *Asteraceae* family members show a wide range of anti-inflammatory, antimicrobial, antioxidant and hepatoprotective activities (Figure 1) [3]. This paper reviews the current state of up-to-date literature concerning the positive effect of plants, particularly the vegetables, from the *Asteraceae* family on human health. 

## 2. Characteristics of the *Asteraceae* Family 

The *Asteraceae* family, often known as the sunflower family, is one of the largest flowering plant families, including over 1600 genera and 25,000 species worldwide. It includes a number of well-known species, such as chicory, sunflower, lettuce, coreopsis, dahlias and daisy, as well as a number of plants of medicinal significance, such as wormwood, chamomile and dandelion [4]. For example, *Carduus* species have often been used as antihemorroidal and cardiotonic remedies in traditional medicine, and *Onopordum tauricum* as a remedy for liver disease. The flowers and roots from *Onopordum acanthium* were used as antipyretic and diuretic agents, and *Centaurea solstitalis* is used in folk medicine in Turkey to treat stomach problems, abdominal pain, herpes infections and the common cold [5]. *Tanacetum parthenium,* also known as feverfew in folk medicine and medieval aspirin, has been used as a remedy for headaches, migraine, nausea, vomiting, stomach-aches, rheumatism and other inflammations [6].

Another plant with practical uses is *Bidens pilosa*, also known as Spanish needles, which grows mostly in subtropical and tropical regions. It has been used as a remedy for liver problems and to lower blood pressure, and is a major ingredient in herbal infusions in Taiwanese folk medicine. In addition, *Carthamus tinctorius* (safflower) is a treatment for rheumatism and osteoporosis in Korean herbal medicine [7], and the juice from *Emilia sonchifolia* roots is used to treat dysentery in Chinese medicine and as a remedy for diarrhea in Nepalese medicine [3]. 

*Cichorium intybus* (chicory) is used in traditional medicine as a remedy for inflammatory inflammation and liver disorders, and is also used to treat gallstones, gout, rheumatism and appetite loss. Tonics from *C. intybus* have also been used to treat enlarged spleen and fever in Indian Ayurveda medicine, and a decoction from leaves was used as a cure for rheumatism and gout [8,9]. 

Many plants from the Asteraceae family have been used in traditional medicine in Turkey. Tea prepared from *Achillea aleppica* and *Achillea biebersteinii* was recommended for abdominal pain. The aerial parts from *Chrysophthalmum montanum* were boiled and applied to wounds and other injuries. The roots were often eaten to reduce high blood pressure. *Matricaria aurea* was recommended in the diet twice a day for bronchitis, sore throat and cough. The seeds of *Notobasis syriaca* were used as remedies for liver disease [10]. 

## 3. Botanically Characteristics of the *Asteraceae* Family

The *Asteraceae* family is widely distributed throughout the world in a variety of ecological habitats, except Antarctica. They are found in forest habitats, high altitude grasslands and even urban green spaces, but they are much less common in tropical areas [11]. The morphology of the *Asteraceae* plants is also diverse. Some species are trees reaching more than 30 m, such as *Dasyphyllum excelsum* in Chile or *Vernonia arborea* in Malaysia; however, many others are shrubs, like rabbit brush or rosette-trees, and most are perennial or less annual herbs, ranging from 1–3 m tall sunflowers and to almost sessile forms. The smallest examples are those of the genus *Mnioides* found in the Peruvian Andes [11]. 

The form of the leaves varies widely: while most are large, others are small and spiny, and some are nonexistent, with their function being taken over by a green stem. Most leaves are covered with an indumentum and hairs of all lengths and colors [11]. Most have a flat cluster of small flowers of various colors. A good example is the Jerusalem artichoke, with thin, yellow flowers on a tall stalk [3,12]. 

## 4. Nutritional Value of *Asteraceae* Family 

Many species of the *Asteraceae* can be included in a regular, healthy diet. A study of the *Asteraceae* by García-Herrera et al. [13] found the protein content to range from 0.4 to 6.13 g per 100 g of edible parts and fiber from 2.55 to 13.44 g. The roots, leaves and flowers are also good sources of Na, K, Ca and Mg, and of vitamins A, B, C and D. Most plants have a low fat content [13]. 

*Crepis vesicaria* and *Sonchus oleraceus* both grow in the Mediterranean area. Both are considered wild edible plants and are often used as additions to salads in Italian cuisine, and both are good sources of vitamin A: 100 g of *C*. *vesicaria* leaves provide 50% of the recommended daily allowance (RDA) of vitamin A and *S. oleraceus* provides over 80%. Additionally, both species contain high levels of thiamine: 200 g of material supplies 15% of the RDA of thiamine. In addition, 200 g of *S. oleraceus* supplies almost 14 mg of lutein per day, which has been associated with a reduction of age-related macular degeneration [14].

*Artemisia absinthium* is used as a flavoring agent in various wines and spirits, and is an important addition to absinthe. *Carthamus tinctorius* (safflower) is especially popular in Portugal, where the seeds are used for cheese manufacture and the leaves are used as food colorants. The young leaves of *Inula crithmoides* are eaten raw, and the fresh shoots can be added to salad or pickled [7].

*Cichorium intybus*, or chicory, contains 22.15 mg of vitamin C per 100 g of dry matter and more than 60% of its total organic acid content is malic acid. Chicory has various uses in the kitchen: the green leaf is a basic ingredient in salads and a popular addition in sandwiches. The roots are used as caffeine-free coffee substitutes. Chicory extracts can be add to nonalcoholic and alcoholic beverages to improve their taste [8,15,16]. Chicory root is a one of the biggest natural sources of inulin. The content of inulin varies from 11–20 g on 100 g of fresh roots and around 44% on dry root weight. The amount of inulin can change depending on season and is the lowest during autumn [17]. 

*Cynara cardunculus* (artichoke), has been consumed for centuries. In ancient times, rich Greeks and Romans consumed immature flowers as high-quality vegetables on special occasions and the mature flowers were used as milk coagulants in cheese production. Nowadays, the flowers are often eaten as frozen and canned delicacies, and are often used for plant-based milk and cheese. The flowers of *Tagetes erecta*, commonly known as the Mexican marigold, are often used as food colorants; they are also added to poultry feed to decrease egg cholesterol level and improve egg yolk pigmentation [18]. 

*Helianthus tuberosus*, Jerusalem artichoke, is also a versatile choice in cuisine. Its edible parts are the tubers, which contain vitamins and minerals such as potassium and phosphorous. It is also a source of inulin, a complex carbohydrate which can promote good health in humans; it is believed that 100 g of Jerusalem artichoke tuber provides almost 10 g of inulin. Inulin increases the absorption of calcium, magnesium and various other minerals. Due to its low calorific value and ability to emulate the texture of traditional fat, it is used as an effective substitute for regular sugar and fat in cookies, cakes and breads. Jerusalem artichoke tubers can be used to enhance the characteristics of fermented milk products: in Canada, their juice is fermented and consumed as a prebiotic drink with blueberry juice [12]. 

## 5. Chemical Characteristics and Health Benefits of Vegetables from the *Asteraceae* Family

Many species of *Asteraceae* demonstrate various pharmacological activities, which have been attributed to their phytochemical components, including essential oils, lignans, saponins, polyphenolic compounds, phenolic acids, sterols and polysaccharides (Figure 2 and Figure 3) [5]. A study of various members of the *Asteraceae* family, viz. *Cirsium arvense*, *Onoporidium acanthium*, *Centaurea solstitailis* and *Carduus acanthoides*, found the total phenolic content extract to range from 8.035 to 90.305 mg GAE/L (milligrams of gallic acid equivalent of plant extract), and total flavonoid content from 18.031 to 185.437 mg QE/L (milligram of quercetin equivalent of plant extract) [5]. 

A wide range of phenolic compounds are found, including chicoric acid, kaempferol and its derivatives, luteolin and its derivatives, quercetin, and apigenin and its derivatives. They are also found in the underground parts of plants; for example, chicory root is a source of many acids such as caffeic acid, chlorogenic acid and isovanillic acid [16,19]. In addition, a number of triterpenes, such as taraxacin, taraxacin acid, fardiol, arnidiol, taraxasterol, α-amiryn and β-amiryn, have been identified in *Taraxacum* spp.: an important member of the family. Many plants are also sources of malic acid, fumaric acid, citric acid and ascorbic acid [20].

Arctiin is a lignan, a glucoside of artigenin, found in many species of *Asteraceae*, particularly *Centaurea imperialis*, *Forsythia viridissima* and *Saussuerea heteromallav* and was first isolated from *Arctium lappa*. Arctiin possesses a number of pharmacological effects including cytotoxicity, antiproliferative and desmutagenic activity; it also acts as a platelet activating factor antagonist and calcium antagonist (Figure 2) [2].

Of the 1100 known acetylenes, i.e., molecules with biological activity, around 200 have been found in the tribes of the *Asteraceae*, including the *Astereae*, *Cynereae*, *Anthemideae* and *Heliantheae*. Each tribe has its own original set of acetylene metabolites, and hence can be used for chemotaxonomy. Although they share the same basic general chemical structure, based on two or more triple bonds, the compounds are diverse and included a range of aliphatic and cyclic structures containing sulfur, nitrogen and oxygen. Acetylenes demonstrate various cytotoxic, anti-inflammatory and antibiotic effects, among others [21].
Figure 3Chemical structure of hydroxycinnamic acid derivatives according to Jaiswal, Kiprotich and Kuhnert [22].
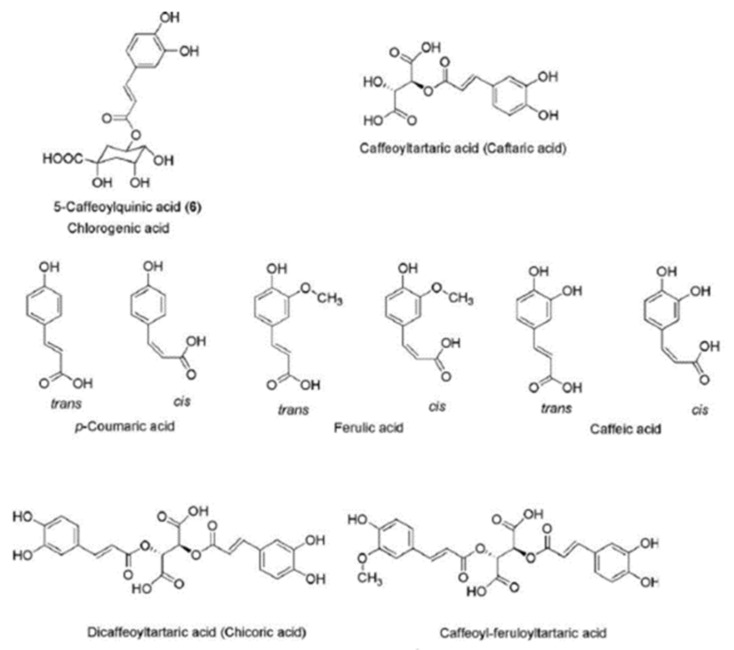


The plants of the *Asteraceae* family are rich sources of chloregenic acid: a hydroxycinnamic acid derivative formed by the reaction between quinic acid and a specific trans-cinnamic acid, such as ferulic, caffeic or *p*-coumaric acid. Chloregenic acids have been found to possess antiviral, antioxidant, antimutagenic, anti-inflammatory and radical-scavenging activities [22]. Various hydroxycinnamic acid derivatives, especially caffeic acid esters, have been found in the leaf (214 mg/g of dry weight) and petal fractions (420 mg/g of dry weight) of dandelion [23].

One of the most important groups of compounds in the *Asteraceae* family is that of the sesquiterpene lactones. They are terpenoids, which represent half of all the sesquiterpenes in the terpenoids group. Their chemical structure is based on a 15-carbon atom skeleton formed around three isoprene units [24]. Sesquiterpene lactones are colorless, have a bitter taste, and are present at particularly high levels in the genera *Vernonia*, *Ambrosia*, *Parthenium* and *Artemisia* [6]. Sesquiterpene lactones promote appetite and digestion due to their bitter taste. Chicory roots are a rich source of sesquiterpene lactones like lactucin, 8-deoxylactucin,13-dihydro-8-deoxylactucin, lactucopicrin,13- dihydrolactucopicrin, jacquinelin, crepidiaside B, lactuside A [17].

*Artemisia absinthium* is a perennial herb commonly known as wormwood. It is often added to biological sprays against pests due to its odor; however, it also demonstrates a range of health properties, including diuretic, digestive, balsamic and depurative effects. It is also recommended as a supplement in leukemia treatment. The aerial part of the plant exhibits snake antivenom activity.

*Erigeron canadensis* shows antiplatelets and anticoagulant activity, especially induced by the cyclooxygenases pathway induced by arachidonic acid. The preparation from this plant can inhibit plasma clot formation in prothrombin time and partial thromboplastin time in human plasma. It has also demonstrated significant anti-IIa activity mediated by cofactor II of heparin [25]. 

*Acmella oleracea* is a source of spilanthol, a compound belonging to N-alkyladmides. It shows diuretic activity and is used in oral health care, often as an addition to toothpaste [7]. Topical application of *Achillea kellalensis* flowers on a wound can hasten healing, due to their flavonoid content. *Achillea millefolium* extract has an estrogenic effect, thanks to its content of phytoestrogens such as apigenin and luteolin; these have a stronger binding affinity to β estrogen receptors than estradiol [3]. *Calendula officinalisis* demonstrates wound healing properties and antibacterial and antiviral activity [8]. 

Many *Asteraceae*, especially Taraxacum spp., Reicardia picroides, Sonchus oleraceus and Picris echioides show bacteriostatic and bacterial potency against Salmonella tymphimurium, Bacillus aureus, Escherichia coli and Staphylococcus aureus. They have also demonstrated antifungal activity against Penicillium ochrochloron (Table 1) [19]. 

Various plants from the *Asteraceae* family demonstrate antimicrobial activity in vitro. An antimicrobial screening assay found ethanol extract from *Ageratum conyzoides* and *Tagetes erecta* to demonstrate antimicrobial properties against a broad spectrum of Gram-positive and Gram-negative bacteria. *T. erecta* was also found to inhibit the growth of *P. aeruginosa* [26]. Chicory also demonstrated antimicrobial effect, due to inhibitory effect on various Gram-positive and Gram-negative bacteria, *Aspergillus niger* and *Sachharomyces cerevisiae* [17].

*Taraxacum officinale*, dandelion, shows strong diuretic activity, probably due to its high potassium content. It can also improve the regenerative capacity of the liver: it was found to suppress monophosphate-activated protein kinase in the livers of mice fed a high-fat diet [20,27]. Lis and Olas [28] reported that dandelion roots demonstrated antiplatelet activity in vitro, based on measurements of acid phosphatase activity during blood platelet adhesion to collagen and fibrinogen; the strongest antiplatelet activity was demonstrated by a fraction with high hydroxyphenylacaetate inositol ester content (Table 1). 

*Cynara cardunculus*, artichoke, has demonstrated hepatoprotective, hypocholesterolemia, hypolipidemic and hypoglycemic properties, which have been attributed to its high phenolic compound content. Artichoke can also serve as a source of dietary prebiotic [18]. 

*Achillea cucullata* is a Turkish and Iranian species with antimicrobial activity. *A. cucullate* extract has been found to inhibit the growth of Gram-positive bacteria like *Staphylococcus aureus* and *Enterococcus faecalis*, and Gram-negative bacteria like *Pseudomonas aeruginosa* and *Escherichia coli* in vitro. It can also inhibit the growth of *Candida albicans* (Table 1) [1]. *Helianthus tuberosus*, Jerusalem artichoke, has demonstrated prebiotic properties, which have been attributed to its inulin content: inulin improves the survival of *Lactobacillus paracasei* BGP1 and *Lactobacillus plantarum* CIDCA8327 strains, and enhances their resistance to gastrointestinal conditions (Table 1) [29].

*Sylibum marianum* is also known as milk thistle. Its major source of silymarin, a mixture of silibinin A and B, silydianin and silychristin. Milk thistle demonstrated various biological activity, including hepatoprotective, cardioprotective and cytoprotective effects. Milk thistle has antidotal and protective effects against numerous biological toxins, like mycotoxin, bacterial toxin and even snake venoms. Silymarin present in milk thistle has shown antioxidant activity against lipid peroxidation induced by aflatoxins. Silymarin also suppressed lipopolysaccharide-induced neuroinflammatory impairment. Beside natural toxins, milk thistle also has a protective effect against various chemical toxic agents, like aluminum, copper, cadmium and lead [30].
ijms-22-03009-t001_Table 1Table 1Various health properties of the *Asteraceae* revealed by in vitro and in vivo experiments.Plants (Preparation/Extract)Chemical Characteristic of Preparation/ExtractType of Research Biological ActivityReferencesExtract from seeds of *H. cretica*, *H. graecum*, *P. echioides*, *R. picroides*, *S. hispanicus*, *S. oleraceus*, *U. picroides* and *T. officinale*α- and β-tocopherols (18.32 and 16.31 µg/100 g fresh weight) oxalic acid (972 mg/100 g fresh weight)In vitroAntimicrobial activity (bacteriostatic and bactericidal potency against *Bacillus areus*, *Salmonella tymphimurium*, *Escherichia coli*, *Penicilium funiculosum*)[19]Extract from Jerusalem artichokeInulin (isolated from roots)In vitro (*L. paracasei* BGP1 and *L. plantarum* CIDCA8327 strain)Prebiotic properties (inulin improved bacterial growth)[29]Aqueous extract from *Achillea cucullata*The total phenol content (53.807 ± 0.059 mg GAE/g dry weight) the total flavonoid content (21.372 ± 0.026 mg QE/g)In vitroAntimicrobial activity (inhibitory effect against *Staphylococcus aureus*, *Pseudomonas aeruginosa*)[1]Extract from the roots from *Taraxacum officinale*
(5 fractions)Hydroxycinnamic acids, hydroxyphenylacetic acid derivatives, sesquiterpene lactones In vitroAntiplatelet activity (inhibitory effect on blood platelet adhesion to endothelial cells)[28]Aqueous extract from *Achillea cucullata*The total phenol content (53.807 ± 0.059 mg GAE/g dry weight) the total flavonoid content (21.372 ± 0.026 mg QE/g)In vitroAntioxidant activity (DDPH free radical scavenging activity)[1]Extract from leaf from *Cichorium intybus*Anthocyanins, (the major–Cyanidin-3-O-(6”-malonyl-β-glucopyranoside))In vitroAntioxidant activity (anthocyanins in leaf have free radical scavenging ability)[9]Extract from leaf and petals from *Taraxacum officinale*Hydroxycitric acids: in the leaf fraction 420 mg/g dry weight (the main component-l-chicoric acid 350 mg/g dry weight); in the petal fraction 214 mg/g dry weightIn vivo (18 male albino Wistar rats)Antioxidant activity (the level of biomarkers of oxidative stress in blood plasma)[23]Extract from *Cynara scolymus*Phenolic acids (mainly chlorogenic acid, cynarin and caffeic acid), sesquiterpene lactones,In vivo (60 male and 60 female Winstar rats)Anti-inflammatory activity (increase in total leukocyte and lymphocyte counts)[31]Extract from *Cichorium intybus*Phenolic acids, sesquiterpene lactones, β-sitosterolIn vivo (6-week-old male mice)Anti-inflammatory activity (increased level of IL-12)[32]

### 5.1. Antioxidant Activity

Extracts from the plants of the *Asteraceae* family demonstrate free radical scavenging ability, which has been attributed to their phenolic compound content. The phenolic compounds act by improving the endogenous antioxidant system, chelating the metal ions and avoiding the formation of free radicals. For example, arctiin has been found to significantly slow increases in intracellular reactive oxygen species (ROS) generation induced by H_2_O_2_: a process which often mediates sudden cell cycle arrest or cell death [2,25,33]. In addition, the lipophilic compounds isolated from feverfew can decrease human neutrophil oxidative burst activity [8]. 

Extracts from *T. officinale* flowers can inhibit supercoiled DNA breakage in vitro induced by hydroxyl and peroxyl radicals, and reduce lipid and protein oxidation in plasma in vitro. The polysaccharide fraction from the roots also appears to improve antioxidant protection mechanisms in an acetaminophen-induced oxidative injury model in mice [20,27]. 

The effect of dandelion on the antioxidant profile of blood plasma and urine samples, and blood plasma lipid level was investigated in vivo. Three groups of six male albino Wistar rats were included in the study. One group was supplemented with dandelion leaf extract, another with dandelion petal extract and a control group which did not receive either. The results indicated a decrease in blood plasma lipid levels and lower oxidative stress in blood plasma, as indicated by thiol group levels and protein carbonylation inhibition (Table 1) [23]. 

Antioxidant activity in fresh chicory leaves was determined by evaluating lipid peroxidation inhibitory activity in vitro using fluorescence spectroscopy and liposome oxidation. The 250 μg/mL leaf extract preparation inhibited 88% of lipid peroxidation. The chromatographic profiles of the plants indicated high levels of anthocyanins, which are known to demonstrate strong antioxidant activity (Table 1) [9]. 

Antioxidant activity is strongly and positively correlated with phenolic content. An in vitro study found high levels of both in the ethyl acetate fraction from Jerusalem artichoke leaves [34]. 

*Silybum marianum* owes its strong antioxidant ability to silibinin (SBN), a flavonolignan isolated from its fruits and seeds. Silibinin demonstrates strong scavenging potential for most free radicals, such as peroxyl radical and hydroxyl radicals. It also inhibits the NF-ĸB pathway by treating and attenuating the inflammatory reaction that stimulated atherosclerosis. In in vivo experiments, SBN has been shown to protect mouse and rat liver against the toxic effects of carbon tetrachloride and alcohol [35]. 

Aqueous ethanolic extract of *Achillea cucullata* has been found to demonstrate antioxidant potential in vitro based on DDPH free radical scavenging assay. The findings indicate an IC_50_ value of 132.55 ± 0.026 μg/mL for the extract, compared to 7.548 ± 0.047 μg/mL for the strong antioxidant gallic acid (Table 1) [1]. 

*Artemisia absinthium* methanolic extract after oral administration at doses of 100 and 200 mg/kg showed scavenging activity on superoxide anion radicals, by restoring superoxide dismutase and glutathione levels and decreasing the level of thiobarbituric acid reactive substances. This leads to the inhibition of oxidative stress caused by cerebral ischemia and reperfusion [25].

### 5.2. Anti-Inflammatory Activity 

*Cynara scolymus*, artichoke, has been found to demonstrate anti-inflammatory activity in vivo in a study of 60 male and 60 female Wistar rats. The animals were treated with 1, 2 or 4 g/kg body weight of *Cynara scolymus* extract for 28 days. Regular treatment with the extract increased total lymphocyte and leukocyte count, interleukin-12 (IL-12) and phagocyte activity, had an immunostimulant effect, as indicated by hemogram, serum biochemistry, lymphoid organ weight, macrophage and neutrophil oxidative burst, and specific humoral immune response (Table 1) [31]. 

The anti-inflammatory activity of 0.1–100 µg/mL *Cichorium intybus*, i.e., chicory, extract was demonstrated in in vivo studies based on six-week-old male C57BL/6 and BALB/c mice. The chicory extract increased production of IL-12 by dendritic cells, i.e., antigen-presenting cells in the immune system (Table 1). Additionally, higher concentrations of extract inhibited allogenic T cell proliferation, but increased the level of IFN-γ at lower concentrations [32]. Chicory extract has also been found to lower the concentration of certain cytokines, such as the anti-inflammatory interleukin-4 [8,34,35]. 

Arctiin plays a crucial role in the anti-inflammatory activities of the *Asteraceae*, due to its ability to inhibit production of inflammatory mediators, including the interleukins IL-6 and IL-1β, prostaglandin E_2_ (PGE_2_), tumor necrosis factors (TNF-α) and nitric oxide. Arctiin also inhibits the translocation pathway of nuclear factor (NF)-kβ, leading to suppression of cyclooxygenase-2 (COX-2) [2]. 

The methanol extract from *Emilia sonchifolia* demonstrates anti-inflammatory effects by inhibition of edema induced by carrageenan [3]. Oleamide isolated from burdock can reduce the production of TNF-α and IL-4 [8]. 

*Taraxacum* species also demonstrate anti-inflammatory activity: extracts from dandelion flowers prevent the production of proinflammatory cytokines like PGE_2_ and suppresses COX-2 and iNOS; in addition, taraxasterol isolated from dandelion inhibits the production of TNF-α, IL-1β, PGE_2_, nitric oxide and IL-6 by preventing NF-kβ translocation in LPS-induced RAW264.7 macrophage models [20].

### 5.3. The Application of Asteraceae in Human Health 

Nowadays, there is an increasing interest in the role of diet in human health and therapy based on natural remedies in the treatments for many ailments. It is proven that a diet rich in plants, the best source of antioxidants, plays a dominant role in preventing these diseases. For example, inulin isolated from dandelion roots is used for microbiological production of a high fructose syrup, as a replacement for the traditional one, and plays a role in the prevention of diabetes and obesity. Coffee from dandelion roots is a great alternative for normal coffee, due to the lack of narcotic effect. In the USA, preparations from dandelion leaves are an addition to health food products and supplements for diuretic problems [28]. Chicory also is a valuable source for new health food products and functional food. The roots from chicory are a healthy replacement for white flour and fat in cracker production, due to a high level of dietary fiber and inulin. They are in addition to various low-calorie sweeteners to increase dietary fiber content [17]. Jerusalem artichoke also is a source of remedies for various diseases. In Russia, the flowers are used for tea, which, used daily, helps to improve the immune system in the body, provides an energy boost and prevents kidney disorders. Tubers of Jerusalem artichoke are recommended in the diet for obesity, as they cause a feeling of satiation [36].

However, further studies the *Asteraceae* family should be conducted to fully understand the potential uses as a prevention for many diseases or in the development of new drugs. 

## 6. Conclusions

The *Asteraceae* family is the most varied and cosmopolitan family of flowering plants. Many of its species have been used in traditional medicine since ancient times. Nowadays, the growing need for more natural sources of medicine has driven scientific interest towards the *Asteraceae* family. Studies have demonstrated that their extracts have a positive impact on human health, thanks to their antioxidant, anti-inflammatory and antimicrobial activities [8]. 

## Figures and Tables

**Figure 1 ijms-22-03009-f001:**
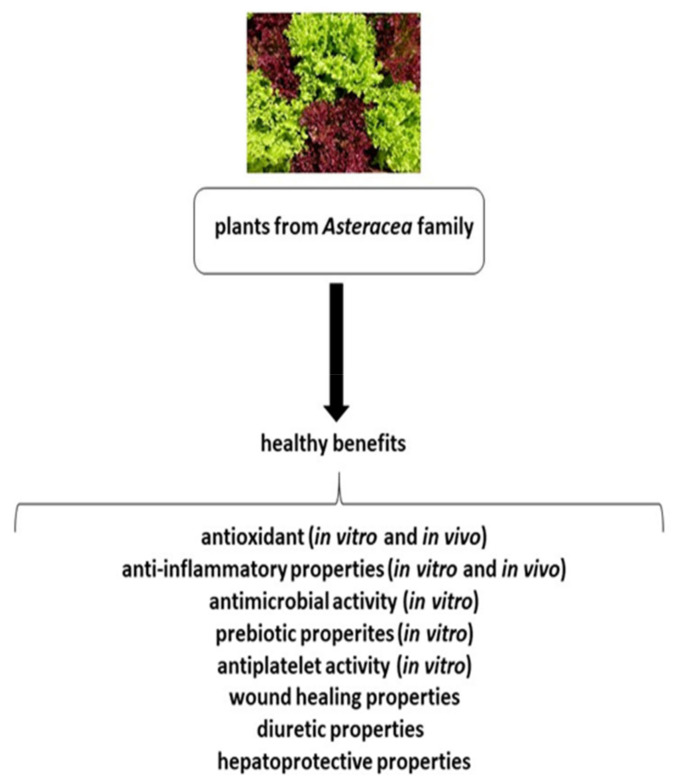
The positive effects of plants from the *Asteraceae* family on human health.

**Figure 2 ijms-22-03009-f002:**
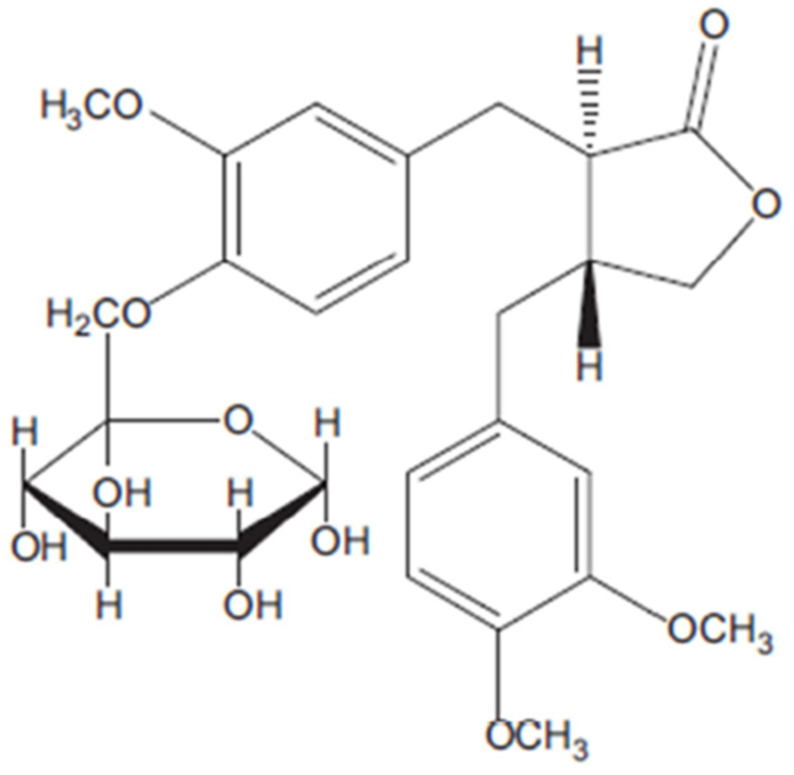
Chemical structure of Arctiin according to Tourchi, Arslan and Iranshahi [2].

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
