# Peer review of "The Plants of the Asteraceae Family as Agents in the Protection of Human Health"

_ijms, 2021, doi:10.3390/ijms22063009_

Round 1

Reviewer 1 Report

page 2: ”antihemrriodal”, correct: antihemorroidal

            ”Onopordum taurium”,   correct: Onopordum tauricum

                ”Tancetum partherium”, correct: Tanacetum parthenium

                ”Bidens Pilosa”, correct:  Bidens pilosa

                ”decocotion”, correct: decoction

Marticaria aurea”, correct: Matricaria aurea

”Veronia arborea”,      correct: Vernonia arborea

” Cynara cardunuculus”, corect: Cynara cardunculus

”Geeks and Romans”, correct: Greeks and Romans

page 3: ”Cantaurea solstitailis”, correct: Centaurea solstitailis

page 4: ”antiviral activity. (”,       correct: antiviral activity (

                ” Many Asteraceae, especially Taraxacum spp., Reicardia picroides,”,                corect: Many Asteraceae, especially Taraxacum spp., Reicardia picroides.....      with italic font 

page 5, 6    ”Achillea cucullate”,          correct: Achillea cucullata

Some species (eg Helianthus tuberosus, Cynara scolymus, Taraxacum) occur several times with the same uses.

The references are placed at the end of the paper, according to figures and tables.

Author Response

page 2: ”antihemrriodal”, correct: antihemorroidal

            ”Onopordum taurium”,   correct: Onopordum tauricum

                ”Tancetum partherium”, correct: Tanacetum parthenium

                ”Bidens Pilosa”, correct:  Bidens pilosa

                ”decocotion”, correct: decoction

Marticaria aurea”, correct: Matricaria aurea

”Veronia arborea”,      correct: Vernonia arborea

” Cynara cardunuculus”, corect: Cynara cardunculus

”Geeks and Romans”, correct: Greeks and Romans

Response: We have corrected all sentences on page 2.

page 3: ”Cantaurea solstitailis”, correct: Centaurea solstitailis

Response: We have corrected.

page 4: ”antiviral activity. (”,       correct: antiviral activity (

Response: We have corrected.

                ” Many Asteraceae, especially Taraxacum spp., Reicardia picroides,”,                corect: Many Asteraceae, especially Taraxacum spp., Reicardia picroides.....      with italic font 

Response: We have corrected.

page 5, 6    ”Achillea cucullate”,          correct: Achillea cucullata

Response: We have corrected.

Some species (eg Helianthus tuberosus, Cynara scolymus, Taraxacum) occur several times with the same uses.

Response: We have corrected it by removing repeating used of the same species.

The references are placed at the end of the paper, according to figures and tables.

Response: I’m sorry, could  you be more specific, because references are placed in the end of manuscript.

Reviewer 2 Report

This is a well-written comprehensive narrative review on updated literature about plants belonging to Asteraceae family and their beneficial effects on human health. Overall, the paper is interesting and provides to the reader an overview of main biological activities of these plants. However, some literature is still missing and should be added:

https://pubmed.ncbi.nlm.nih.gov/32768902/

https://pubmed.ncbi.nlm.nih.gov/30033764/

https://www.frontiersin.org/articles/10.3389/fphar.2020.00852/full

Besides the references above, the search for articles can be further improved by the Authors.

Since the main topic of the review is “health benefits”, I would suggest to add a paragraph on the evidence about the effects, in terms of prevention and/or (adjuvant) treatment, of these plants on human diseases (such as cardiovascular diseases), not only focusing just on biological activities.

 I would suggest to enrich the review by adding a figure on the chemical structure of main phytochemical components found in many species of Asteraceae.

Author Response

This is a well-written comprehensive narrative review on updated literature about plants belonging to Asteraceae family and their beneficial effects on human health. Overall, the paper is interesting and provides to the reader an overview of main biological activities of these plants. However, some literature is still missing and should be added:

https://pubmed.ncbi.nlm.nih.gov/32768902/

https://pubmed.ncbi.nlm.nih.gov/30033764/

https://www.frontiersin.org/articles/10.3389/fphar.2020.00852/full

Besides the references above, the search for articles can be further improved by the Authors.

Response: We have added the references:

“Erigeron canadensis show antiplatelets and anticoagulant activity, especially by cyclooxygenases pathway induced by arachidonic acid. The preparation inhibited plasma clot formation in prothrombin time and partial thromboplastin time in human plasma. It also demonstrated significant anti-IIa activity mediated by cofactor II of heparin (Michel, Abd Rani and Husain 2020).”

“Artemisia absinthium methanolic extract after oral administration at dose 100 and 200 mg/kg showed scavenging activity on superoxide anion radicals, by restoring superoxide dismutase and glutathione level and decreasing  level of thiobarbituric acid reactive substances. This lead to inhibition of oxidative stress cause by cerebral ischemia and reperfusion (Michel, Abd Rani and Husain 2020).”

“Sylibum marianum is also known as milk thistle. Its major source of silymarin, a mixture of silibinin A and B, silydianin and silychristin. Milk thistle demonstrated various biological activity including hepatoprotective, cardioprotective and cytoprotective effects. Milk thistle have antidotal and protective effect against numerous biological toxins, like mycotoxin, bacterial toxin and even snake venoms. Silymarin present in milk thistle showed antioxidant activity against lipid peroxidation induced by aflatoxins. Silymarin also suppressed lipopolysaccharide-induced neuroinflammatory impairment. Beside natural toxin milk thistle have protective effect against various chemical toxic agents, like aluminum, copper, cadmium and lead (Fanoudi et al. 2020).”

“Sesquiterpene lactones promotes appetite and digestion due to their bitter taste. Chicory roots are a rich source of sesquiterpene lactones like lactucin, 8-deoxylactucin,13-dihydro-8-deoxylactucin, lactucopicrin,13- dihydrolactucopicrin, jacquinelin, crepidiaside B, lactuside A (Perović et al. 2021).”

“Chicory also demonstrated antimicrobial effect, duets inhibitory effect on various Gram-positive and Gram-negative bacteria, Aspergillus niger and Sachharomyces cerevisiae (Perović et al. 2021).”

“Chicory roots is a one of the biggest natural source of inulin, a non-digestible carbohydrate belonging to fructans. The content of inulin varies from 11-20 g on 100 g of fresh roots and around 44% on dry roots weight. The amount of inulin can change depending on season and is the lowest during amount (Perović et al. 2021).” 

Since the main topic of the review is “health benefits”, I would suggest to add a paragraph on the evidence about the effects, in terms of prevention and/or (adjuvant) treatment, of these plants on human diseases (such as cardiovascular diseases), not only focusing just on biological activities.

Response: We have added suggested paragraph.

“5.3 The application of Asteraceae in human health

Nowdays there is an increasing interest in the role of diet in human health and therapy based on natural remedies in treatment for many ailments. It is proven that diet rich in plants, the best source of antioxidants, plays a dominant role in preventing these diseases.  For example, inulin isolated from dandelion roots is used for microbiological production of high fructose syrup, as replacement for traditional one, and plays a role in prevention of diabetes and obesity. Coffee from dandelion roots is a great alternative for normal coffee, due to lack of narcotic effect. In the USA preparations from dandelion leaves are addition to health food products and supplements for diuretic problems  (Lis and Olas 2019). Chicory also is a valuable source for new health food products and functional food. Roots from chicory are healthy replacement for white flour and  fat in cracker production, due to high level of dietary fiber and inulin. They are addition to various low-calorie sweetener to increase dietary fiber content  (Perović et al. 2021). Jerusalem artichoke also is a source of remedies for various diseases. In Russia, flowers are used for tea, that daily used helps improve immune system in the body, provide energy boost and prevent kidney disorders. Tubers of Jerusalem artichoke are recommended in diet for obesity, as they cause feeling of satiation (Sawicka et al. 2020).”

However, further studies on Asteraceae family should be conducted to fully understand the potential used as a prevention for many diseases or in development of new drugs.

I would suggest to enrich the review by adding a figure on the chemical structure of main phytochemical components found in many species of Asteraceae.

Response: We have added figures of the chemical structure of phytochemical components

Round 2

Reviewer 1 Report

Question: Authors quoted in the text, should not be marked with numbers, according to the instructions, according to https://www.mdpi.com/journal/ijms/instructions?

row 2 ”antihemrriodal”, correct: antihemorroidal

The References are placed at the end of the paper.

Author Response

Question: Authors quoted in the text, should not be marked with numbers, according to the instructions, according to https://www.mdpi.com/journal/ijms/instructions?

Response: We have corrected.

row 2 ”antihemrriodal”, correct: antihemorroidal

Response: We have corrected.

The References are placed at the end of the paper.

Response: We have corrected.